# LncRNA 8244-ssc-miR-320-CCR7 Regulates IFN-β during SVA Infecting PK-15 Cells

**DOI:** 10.3390/microorganisms11030688

**Published:** 2023-03-08

**Authors:** Xiaoyu Tang, Ruiyu Zhang, Long Gao, Xiaocheng Lv, Yuan Sun, Jingyun Ma

**Affiliations:** 1Guangdong Provincial Key Laboratory of Agro-Animal Genomics and Molecular Breeding, College of Animal Science, South China Agricultural University, Guangzhou 510642, China; 2Guangdong Laboratory for Lingnan Modern Agriculture, Guangzhou 510642, China

**Keywords:** Senecavirus A, lncRNA 8244, ssc-miR-320, CCR7, TLR signaling pathway, IFN-β

## Abstract

Seneca Valley virus (SVV), a member of the Picornaviridae family, is an oncolytic RNA virus that can cause idiopathic vesicular disease and increase mortality in newborn piglets. Although research on the pathogenic characteristics, epidemiology, pathogenic mechanism, and clinical diagnosis of SVA has increased due to its emergence and prevalence, the interaction between SVA and its host lncRNA has not been fully studied. This study used qualcomm sequencing to analyze differentially expressed lncRNAs and found that during SVA infection, lncRNA 8244 was significantly down-regulated in both PK-15 cells and piglets. Further analysis through quantitative real-time PCR and dual luciferase experiments demonstrated that lncRNA8244 could compete with ssc-miR-320 to regulate the expression of CCR7. The lncRNA824-ssc-miR-320-CCR7 axis activated the TLR-mediated signaling pathway, which recognized viral molecules and induced the expression of IFN-β. These findings provide new insight into the interaction between lncRNA and SVA infection, which could lead to a better understanding of SVA pathogenesis and contribute to the prevention and control of SVA disease.

## 1. Introduction

Seneca Valley virus (SVV), referred to as Seneca virus (Senecavirus A, SVA), and classified as a small RNA virus within the Picornaviridae family’s Senecavirus genus, was first reported in Brazil in late 2014 [1,2]. Since 2015, it has rapidly spread across various regions and countries, infecting pigs of different ages [3,4,5]. In 2017, both the United States and China reported new cases of SVA infection, and the year of 2016 was seen as a pivotal moment in China’s SVA epidemic. Prior to 2016, the SVA strains that surfaced in China had a higher nucleotide homology with those isolated in Brazil and Canada, whereas strains reported after 2016 had greater similarity with those from the United States [6,7]. Using TaqMan-based quantitative reverse transcription PCR (qRT-PCR) analysis, researchers have identified SVA in various tissues of newborn piglets, including the lung, heart, bladder, kidney, spleen, tonsils, and small intestine, indicating that these tissues can be examined for SVA [2,8,9]. Studies have also shown that SVA causes multiple organ diseases in piglets and leads to multi-system diseases in infected piglets. Despite considerable progress in SVA-related research, there is still a dearth of knowledge, particularly about the interaction between the virus and host lncRNAs.

The lncRNA is a non-coding RNA molecule that is longer than 200 nucleotides and cannot produce proteins [10]. In mammals, thousands of lncRNAs have been identified to be involved in various cellular processes, including metabolism, apoptosis, proliferation, and innate immunity [11,12,13,14]. Specifically, lncRNA acts as a regulator on a variety of key molecules or links, such as pathogen-recognition-receptor-related signals, transcription factor translocation and activation, interferon (IFN) and cytokine production, interferon-activated JAK-STAT signal and antiviral ISG transcription, thus playing an important role in the natural immune response [15]. For example, lnc-Lsm3b induced by IFN can competitively bind to the RIG-I monomer with viral RNA, and feedback inactivation of the innate function of RIG-I in the late stage of the innate reaction. This binding restricts the conformational shift of RIG-1 protein, prevents downstream signal transduction, and terminates the production of type I IFN [16]. Our study focuses on ssc-miR-320 and CCR7, which have been reported to regulate various cellular functions, including cell proliferation, invasion, apoptosis, and glucose and lipid metabolism disorder-associated diseases [17,18]. CCR7 plays a pivotal role in regulating tissue immunity and inflammation through the CCL19-CCR7 axis. It can regulate DC morphologic change, thymic T cell development, and suppress DC apoptosis, leading to the regulation of adaptive immunity and tolerance [17,18]. This study explored the role of the lncRNA 8244-ssc-miR-320-CCR7 axis.

SVA is an emerging porcine blistering pathogen that causes porcine blistering disease outbreaks in different regions. However, knowledge about SVA immunity remains limited. To address this gap, we conducted a study using high-throughput sequencing to identify differentially expressed lncRNAs following SVA infection of PK-15 cells. Our objective was to investigate the targeting relationship between lncRNA 8244-ssc-miR-320-CCR7 and its regulatory role on key molecules of the TLR signaling pathway and IFN-β, with the aim of revealing how SVA regulates host immune responses by manipulating lncRNA expression. The results of our research will provide important theoretical and practical guidance for establishing disease prevention and control strategies for SVA. Specifically, our findings will help to establish a molecular strategy for immune escape through the regulation of lncRNA expression.

## 2. Materials and Methods

### 2.1. Cells, Viruses and Clinical Samples of Piglets

The cell line (Porcine kidney cells (PK-15)) used in this experiment was obtained from the Poultry Research Laboratory, School of Animal Science, South China Agricultural University, while the SVA CH-01-2015 isolate was isolated and preserved by our laboratory.

Huanong Wen’s Co., Ltd. (Yunfu, China) provided clinical samples of piglets, including two samples each of small intestine, kidney, lung, and brain.

### 2.2. Plasmids and Antibodies

The pmirGLO vector (genepharma) for dual luciferase experiments is kept in our laboratory. PmirGLO-wild-CCR7 and pmirGLO-mut-CCR7 were constructed based on the binding site of CCR7 and miRNA-320. PmirGLO-wild-8244 and pmirGLO-mut-8244 were constructed based on the binding site of lncRNA 8244 and miRNA-320. These plasmids are utilized for verifying targeting relationships. The plasmid expressing the CCR7 gene (pmCherry-CCR7) was purchased from the miaoling plasmid platform (Miaolingbio, China). Sus scrofa_NONSUSG008244.1 (http://www.noncode.org, accessed on 1 February 2023) overexpression lentiviral vector was constructed by Jiman Biotechnology Co., Ltd. (Zhaoqing, China). Additionally, miRNA-320 mimics and inhibitor were purchased from Jiman Biotechnology Co., Ltd., while anti-β-actin (rabbit) monoclonal antibody (abcam), anti-TLR3 (pig) monoclonal antibody (abcam), mouse IFN-beta antibody (R&D). Anti-SVA-VP1 are kept in our laboratory.

### 2.3. Quantitative Real-Time PCR (qPCR)

Total RNA was isolated from cells with TRIzol reagent (Invitrogen, Waltham, MA, USA). Portions (1 μg) of each RNA sample were reverse-transcribed by using a PrimeScript RT reagent kit (TaKaRa, Kusatsu, Japan), and cDNA was quantified with these genes’ specific primer pairs (Table 1). Primers were designed to target conserved regions of genes using primer-Premier 5.0 (Premier Biosoft Interpairs, Palo Alto, CA, USA). qPCRs were conducted in the ABI7900HT (Applied Biosystems, Waltham, MA, USA) using a SYBR Premix Ex Taq kit (TaKaRa), and reactions were denatured at 95 °C for 30 s, followed by 40 two-step cycles of 95 °C for 5 s and 60 °C for 30 s. The relative gene expression levels were normalized against that of GAPDH (glyceraldehyde-3-phosphate dehydrogenase). The specific primer sequence is shown in Table 1.

### 2.4. Cell Infection

Cells with 80–90% confluence in good growth condition were aspirated, washed twice with PBS, spliced with SVACH-01-2015 (MOI = 1.5), and incubated in an incubator for 1 h. Finally, culture medium containing 2% fetal bovine serum was added and placed in the incubator at 37 °C with 5% CO_2_.

### 2.5. Transduction and Western Blotting

The total cell protein was extracted 24 h after transfection. Cells were collected and treated with lysis buffer (Roche, UK). Protein concentrations of whole cell lysates were determined using a bicinchoninic acid protein assay kit (Thermo Scientific, Waltham, MA, USA) to assess protein expression. Equal amounts of proteins were separated using 12% sodium dodecyl sulfate-polyacrylamide gels (SDS-PAGE) and transferred to polyvinylidene difluoride (PVDF) membranes (Roche, UK). These membranes were blocked with 5% skim milk in 1× tris-buffered saline (TBS) plus for 2 h at room temperature. Subsequently incubated with diluted primary antibody for 2 at room temperature. Anti-rabbit or anti-pig IgG antibodies conjugated to horseradish peroxidase (HRP) were used as secondary antibodies. Enhanced chemiluminescent substrates were used for detection using the HRP kit (Thermo Scientific, USA).

### 2.6. Dual Luciferase Activity Detection

When the cell confluence reached 60–80%, the transfection mixture was added dropwise to a cell culture plate containing 2% fetal calf serum, shaken gently, and the medium was changed 4 h after transfection as needed.

PK-15 cells (1 × 10^5^) were inoculated in 24-well plates 24 h prior to transfection. Cells were cotransfected with the reporter plasmids encoding lncRNA 8244 and CCR7 and the desired expression plasmid. The empty vector was used as a negative control to adjust the total amount of transfected DNA. A dual luciferase reporter assay system (E1910; Promega, Madison, WI, USA) was used, following the manufacturer’s instructions. All reporter gene assays were repeated six times.

### 2.7. Statistical Analysis

The various treatments were compared using an unpaired, two-tailed Student t test with an assumption of unequal variance. Data are expressed as the mean ± standard deviation from at least three independent experiments. SPSS 17.0 software package (SPSS, Chicago, IL, USA) was used to analyze the qRT-PCR data. *p* < 0.05 was considered statistically significant.

## 3. Results

### 3.1. SVA Infection Inhibits the Expression of lncRNA 8244

To investigate whether the differential expression of lncRNA 8244 is related to SVA infection concentration, PK-15 cells were exposed to various titers of SVA. As the MOI increased, the expression of lncRNA 8244 was suppressed compared to uninfected cells, with an MOI of about 0.3 PFU/cell resulting in a significant decrease in lncRNA 8244 expression (*p* < 0.01) (Figure 1A). At an MOI of 1.5, PK-15 cells were infected with SVA, and quantitative real-time PCR (qPCR) analysis indicated a significant down-regulation of lncRNA 8244 expression at 12 h postinfection (hpi). At 24 hpi, lncRNA 8244 exhibited the lowest expression level (*p* < 0.05) (Figure 1B). In addition, PK-15 cells were infected with pseudorabies virus (PRV) and swine acute diarrhea syndrome coronavirus (SADS-CoV) at an MOI of approximately 1.5 PFU/cell, and the expression of lncRNA 8244 was significantly down-regulated, suggesting that the differential expression of lncRNA 8244 is related to virus infection (Figure 1C). We also detected the expression of lncRNA 8244 in clinical samples from SVA-infected piglets and cell, and the results showed a significant decrease in lncRNA 8244 expression in the small intestine, kidney, lung, brain and PK-15 cell of infected piglets compared to uninfected piglets, suggesting that SVA infection may inhibit the expression of lncRNA 8244 (Figure 1D and Appendix A).

### 3.2. LncRNA 8244 Promotes the Production of IFN-β after SVA Infection in PK-15 Cells

To investigate the regulatory effect of lncRNA 8244 on IFN-β expression, PK-15 cells were transfected with 2.0 μg of plasmid 8244/plasmid NC and 80 pmol of iNC/si-lncRNA 8244. After 24 h, the cells were infected with SVA at a 1.5 MOI and harvested at 24 hpi. The results demonstrated that both mRNA and protein levels of IFN-β expression were significantly higher in the plasmid 8244 group compared to the control group, while IFN-β expression was significantly suppressed in the si-lncRNA 8244 group (*p* < 0.01) (Figure 2A,B and Appendix A). These findings suggest that lncRNA 8244 positively regulates IFN-β expression in SVA-infected PK-15 cells.

### 3.3. LncRNA8244 Promotes the Expression of Key Signal Molecules in the TLR Signaling Pathway

To explore the impact of lncRNA 8244 on the signal pathways related to natural immunity, nucleic acids were extracted for quantitative real-time PCR analysis. We found that the expression of key molecules in the TLR signaling pathway was significantly up-regulated after the cell transfection with plasmids, with TLR3 molecules showing an extremely significant up-regulation (*p* < 0.01) (Figure 3A). Further analysis of the regulatory effect of lncRNA 8244 on the TLR3 signaling pathway revealed a significant up-regulation of TRIF expression in PK-15 cells transfected with plasmid 8244 at 24 h, 48 h, and 72 h (*p* < 0.01). Similarly, TBK1 expression was significantly up-regulated after 72 h of inoculation (*p* < 0.01). However, the expression of TRAF3 and IRF3 did not show significant differences from those in the control group (*p* ≥ 0.05) (Figure 3B). These findings suggest that lncRNA 8244 affects the signal pathway involved in TLR3. To explore the regulatory effect of lncRNA 8244 on TLR3 during SVA infection, fluorescence quantitative experiments and Western Blot experiments were performed. The results revealed that transfection with plasmid 8244 significantly promoted the expression of TLR3 in PK-15 cells infected with SVA (*p* < 0.01). TLR3 expression was also significantly upregulated in clinical samples (Appendix A). Conversely, transfection with si-lncRNA 8244 significantly inhibited TLR3 expression compared to the control group (*p* < 0.05) (Figure 3C,D and Appendix A). Taken together, these findings indicate that lncRNA 8244 promotes the expression of TLR3.

### 3.4. LncRNA 8244 Targets miR-320

The software RegRNA2.0 [19] and the Miranda algorithm were utilized to predict miRNAs that could target lncRNA 8244. After evaluating the predicted scores and complementary matching free energy, it was determined that ssc-miR-320 was the target miRNA of lncRNA 8244 (Figure 4A). To validate this prediction, nucleic acid samples were collected, and a fluorescence quantitative experiment was conducted to measure the expression of lncRNA 8244. The pmirGLO-8244 and pmirGLO-mut-8244 plasmids were constructed according to the binding sites. The results showed that ssc-miR-320 mimics and inhibitors had the expected effects (Figure 4B). The expression of lncRNA 8244 was significantly reduced in the ssc-miR-320 mimics group compared to the NC group (*p* < 0.05). Conversely, the expression of lncRNA 8244 was significantly increased in the ssc-miR-320 inhibitor group compared to the iNC group in SVA-infected cells (*p* < 0.05) (Figure 4C). The targeting relationship was further confirmed through a dual luciferase experiment. The relative luciferase activity of the ssc-miR-320 mimics group was significantly lower than that of the NC group after transfection of the lncRNA 8244 wild-type plasmid (*p* < 0.05). However, there was no significant difference in the relative dual luciferase activity of the ssc-miR-320 mimics group after transfection with the lncRNA 8244 mutant plasmid compared to the NC group (*p* ≥ 0.05) (Figure 4D). Furthermore, the expression of ssc-miR-320 was significantly upregulated in SVA-infected cells and clinical samples (*p* < 0.05). In clinical samples and cell, ssc-miR-320 expression was significantly increased in the kidney, lung, brain and PK-15 cell of SVA-infected piglets compared to normal piglets (*p* < 0.05) (Figure 4E and Appendix A). Overall, the results confirm that ssc-miR-320 is the target miRNA of lncRNA 8244, and there is a negative target relationship between them.

### 3.5. LncRNA 8244 Affects IFN-β Secretion by Regulating the Expression of ssc-miR-320 in PK-15 Cells Infected by SVA

PK-15 cells were subjected to transfection with NC, ssc-miR-320 mimics, iNC, or ssc-miR-320 inhibitor, along with the co-transfection of plasmids NC and NC, plasmid 8244 and ssc-miR-320 mimics, or si-lncRNA 8244 and ssc-miR-320 inhibitor. The cells were then infected with SVA at a 1.5 MOI. The results indicated that compared to the control group, the expression of IFN-β was significantly down-regulated in the ssc-miR-320 mimics transfection group (*p* < 0.05), while it was significantly up-regulated in the ssc-miR-320 inhibitor transfection group (*p* < 0.05) (Figure 5A). Additionally, the co-transfection group with plasmid 8244 and ssc-miR-320 mimics or si-lncRNA 8244 and ssc-miR-320 inhibitor showed no significant difference in IFN-β expression levels compared to the control group (*p* ≥ 0.05) (Figure 5B,C and Appendix A). This suggests that ssc-miR-320 counteracted the effect of lncRNA 8244 on IFN-β expression. The Western blot experiment results corroborated the fluorescence quantification results, indicating that lncRNA 8244 modulates ssc-miR-320, thereby regulating the expression of IFN-β.

### 3.6. LncRNA 8244 Affects the Expression of Key Signal Molecules in TLR3-Related Signaling Pathways by Regulating the Expression of ssc-miR-320

LncRNA 8244 has been shown to impact the expression of key signaling molecules in the TLR3 signaling pathway, with ssc-miR-320 being its target gene. To explore whether lncRNA 8244 acts through ssc-miR-320, the study examined the expression of key signaling molecules in the TLR3 pathway. The results showed that compared to the control group, transfection with ssc-miR-320 mimics resulted in significant down-regulation of TRIF and TBK1 expression 24 h after inoculation (*p* < 0.05), and TRAF3 expression was significantly reduced 48 h after inoculation (*p* < 0.01) (Figure 6A,B). Building on the previous findings that lncRNA 8244 affects IFN-β expression through ssc-miR-320, the study investigated whether lncRNA 8244 regulates TLR3 by modulating ssc-miR-320 expression. Co-transfection of plasmid 8244 and ssc-miR-320 mimics did not significantly differ from the control group in TLR3 expression. Similarly, co-transfection of si-lncRNA 8244 and ssc-miR-320 inhibitor did not significantly differ from the control group in TLR3 expression. The expression of ssc-miR-320 counteracted the effect of lncRNA 8244, indicating that lncRNA 8244 regulates TLR3 expression by modulating ssc-miR-320 expression (Figure 6C,D and Appendix A).

### 3.7. CCR7 Was the Target Gene of ssc-miR-320

RegRNA2.0 and the Miranda algorithm, two online target gene prediction software packages, predicted that CCR7 is a target gene of ssc-miR-320. Subsequent experiments confirmed the negative regulatory relationship between ssc-miR-320 and CCR7 (*p* < 0.01) (Figure 7A,B). To further validate this targeting relationship, a dual luciferase experiment was performed. After transfecting the pmirGLO-wild-CCR7 plasmid for 24 h, the ssc-miR-320 mimics group showed significantly lower relative luciferase activity than the NC group (*p* < 0.01). When the pmirGLO-mut-CCR7 plasmid was transfected, the relative dual luciferase activity of the ssc-miR-320 mimics group was not significantly different from the NC group. These results confirm that CCR7 is the targeted mRNA of ssc-miR-320 (Figure 7C).

Simultaneously, PK-15 cells were transfected with iNC/s-lncRNA8244, plasmid NC/plasmid 8244, and co-transfection of plasmid 8244 and ssc-miR-320 mimics, si-lncRNA8244 and ssc-miR-320 inhibitor, and the expression of CCR7 was examined. It was observed that CCR7 expression was significantly increased in the plasma 8244 transfection group compared to the control group (*p* < 0.01), while the si-lncRNA8244 transfection group exhibited a significant decrease in CCR7 expression compared to the control group (*p* ≥ 0.05). The co-transfection group showed no significant difference in CCR7 expression compared to the control group (*p* ≥ 0.05) (Figure 7D). Furthermore, the expression of CCR7 was detected in SVA-infected cells and clinical samples, and it was found to be significantly up-regulated compared to the control group (*p* < 0.05) (Figure 7E and Appendix A). These results indicate that lncRNA 8244 functions as a competitive endogenous RNA by competitively binding to ssc-miR-320, and indirectly regulates the expression of CCR7.

### 3.8. After SVA Infection, the lncRNA 8244-ssc-miR-320-CCR7 Axis Can Regulate the Production of IFN-β by PK-15 Cells

This study investigated whether the ssc-miR-320-mediated regulation of CCR7 affects the expression of IFN-β during SVA infection in PK-15 cells. The efficiency of plasma CCR7 expression and si-CCR7 interference was confirmed through experiments, with plasmid CCR7 and si-CCR7 performing their respective functions (Figure 8A). After transfecting iNC/si-CCR7/plasmid NC/plasmid CCR7 into PK-15 cells, the cells were infected with SVA at a 1 MOI and harvested 24 h post-infection. The results revealed a positive regulatory relationship between CCR7 and IFN-β.

This study investigated the regulatory relationship between ssc-miR-320, CCR7, and IFN-β during SVA infection in PK-15 cells. The results confirmed that CCR7 is a target gene of ssc-miR-320 and can positively regulate IFN-β expression. The study identified a regulatory axis of lncRNA 8244-ssc-miR-320-CCR7 and demonstrated that lncRNA 8244 indirectly regulates IFN-β expression through competitive binding to ssc-miR-320 and the subsequent regulation of CCR7 expression. Further experiments were conducted to verify the regulatory effect of each component of the axis on IFN-β expression. Si-CCR7 reversed the promotion of lncRNA 8244 on IFN-β, and CCR7 reversed the down-regulation of si-lncRNA 8244 on IFN-β. The results from Western blot and ELISA assays were consistent with quantitative real-time PCR (Figure 8B,C and Appendix A). These findings suggest that lncRNA 8244, ssc-miR-320, and CCR7 have a significant impact on IFN-β expression during SVA infection.

### 3.9. The LncRNA 8244-ssc-miR-320-CCR7 Axis Regulates the Expression of Key Signal Molecules in TLR3 Involved the Signaling Pathway

This study revealed that lncRNA 8244 modulates the expression of IFN-β by competitively binding to ssc-miR-320, which indirectly impacts the expression of CCR7 and alters the expression of key molecules in the TLR signaling pathway following transfection with plasmid CCR7. Notably, the levels of TLR3 and TLR8 were up-regulated (Figure 9A). Furthermore, the expression of TLR3 signaling pathway molecules was found to be up-regulated in the transfection group compared to the control group at 24 hpi (*p* ≥ 0.05), including TLR3, TBK1, and TRIF (Figure 9B). LncRNA 8244 and ssc-miR-320 were found to regulate the expression of TLR3. Additionally, a positive regulatory relationship between CCR7 and TLR3 was observed. However, there was no significant difference in TLR3 expression between the co-transfected plasmid 8244 and si-CCR7 group and the control group (*p* ≥ 0.05) (Figure 9C,D and Appendix A). LncRNA 8244 indirectly modulates the expression of CCR7, inhibiting the biological activity of ssc-miR-320, which ultimately affects the regulation of CCR7 on TLR3.

## 4. Discussion

SVA is known to cause vesicular disease, which presents similar clinical symptoms to other vesicular virus diseases such as foot-and-mouth disease (FMD), vesicular stomatitis (VS), and swine vesicular disease (SVD) [20]. Recent research has shown that long non-coding RNAs (lncRNAs) play a significant role in the interaction between viruses and hosts [21]. In this study, we investigated the interaction between SVA infection and host lncRNA for the first time. Our results demonstrate that lncRNA 8244 can up-regulate the expression of IFN-β through the LncRNA 8244-ssc-miR-320-CCR7 axis following SVA infection. Furthermore, we found that lncRNA 8244 modulates the expression of IFN-β through the TLR signaling pathway.

LncRNA has been shown to participate in IFN-mediated innate immune regulation through histone modulation and the regulation of IFN or ISG expression [22,23]. At the transcriptional level, lnc-Lsm3b is a host-derived lncRNA that induces IFN-β production and competes with viral RNA to inactivate RIG-I function [16]. Our previous research showed that IFN-β expression increased after SVA infection, and this study aimed to investigate why. We found that the expression of lncRNA 8244 was down-regulated after SVA infection, and the overexpression of lncRNA 8244 resulted in increased IFN-β mRNA and protein expression levels at 24 hpi. Recent studies have shown that lncRNAs act as competitive endogenous RNA by binding miRNA with mRNA, reducing free miRNA content and regulating mRNA function [24,25]. Dual luciferase experiments showed a negative regulatory relationship between lncRNA 8244 and ssc-miR-320. Our reversal test showed that miR-320 expression can reverse the regulation of IFN-β by lncRNA 8244. The downstream target gene of ssc-miR-320 involved in the regulation of IFN-β was found to be CCR7, with a negative regulatory relationship. In the clinical sample assay, we found that the expression of lncRNA 8244 was significantly down-regulated in the small intestine, but the expression of miRNA-320 and CCR7 was close to that of the control group. The reason is that the expression profiles of CCR7 as well as miR-320 were initially determined by high-throughput sequencing results of SVA-infected PK-15 cell, so there is some tissue variability. We discovered a positive regulatory relationship between CCR7 and IFN-β, and CCR7 can reverse the regulation of IFN-β by lncRNA 8244. Previous studies have shown that CCR7 down-regulates IFN-γ and IFN-γ response genes, enabling B16 cells to evade immunity [26].

The innate immune response serves as the primary defense mechanism against viral invasion, relying on pattern recognition receptors (PRRs) to detect pathogen-associated molecular patterns (PAMPs) [27]. Activation of the signal pathway via PRRs leads to the production of type I interferons and pro-inflammatory factors, with RLR, TLR, and NLR-mediated pathways being the most common [28]. This study focuses on the lncRNA 8244-ssc-miR-320-CCR7 axis and its regulation of IFN-β expression, examining how the axis influences the TLR signaling pathway to modulate IFN-β production. Results revealed that lncRNA 8244, CCR7, and ssc-miR-320 can affect the expression of signal transduction molecules within the TLR pathway. Host cell response to viral infections hinges on PRRs, with Toll-like receptors (TLRs) being the most prominent. LncRNA 8244 and CCR7 significantly up-regulate the expression of TLR3, whereas ssc-miR-320 down-regulates its expression. TLR3-mediated signaling pathways play a crucial role in the antiviral response to poliovirus (PV) infection, as TLR3-deficient mice fail to produce IFN in their serum [29]. Results showed that CCR7 reverses the regulation of TLR3 by lncRNA 8244, indicating that the lncRNA 8244-ssc-miR-320-CCR7 axis activates the TLR-mediated signaling pathway, with TLR3 recognizing viral molecules and inducing the expression of IFN-β. TLR3 induces IFN-β expression via a MyD88-independent pathway, activating IRF3, a key transcription factor responsible for IFN gene induction [30]. The protease activity of SVA 3Cpro can mediate the degradation of IRF3 and IRF7, reducing IFN-β, IFN-α1, IFN-α4, and ISG54 mRNA expression induced by these factors [31,32]. The current study highlights how the lncRNA 8244-ssc-miR-320-CCR7 axis regulates IFN-β expression by modulating the TLR3-mediated signaling pathway in response to SVA infection. Further studies are needed to gain a better understanding of the mechanisms underlying the axis’s effect on IFN-β expression from TLR3 to IFN-β.

## Figures and Tables

**Figure 1 microorganisms-11-00688-f001:**
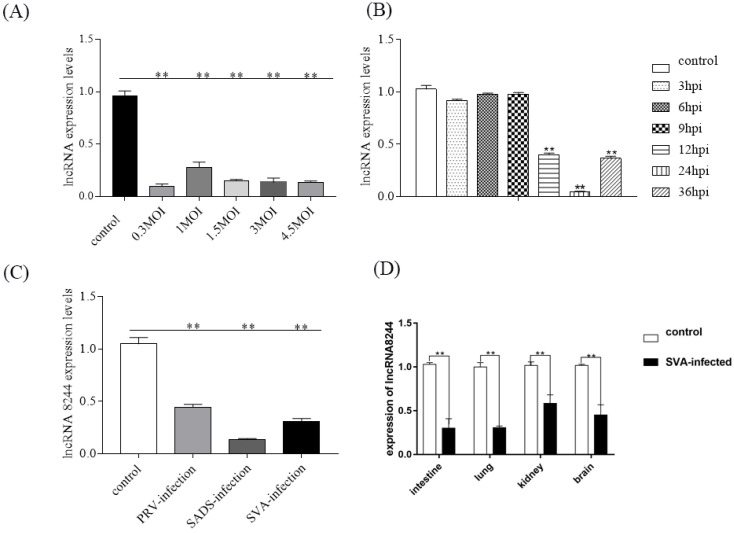
The expression of lncRNA 8244 in cells or clinical samples. (**A**) PK-15 cells were infected or no-infected with SVA (MOI = 0.3, 1, 1.5, 3, 4.5) for 24 h. Total RNA was extracted to determine the relative mRNA expression of lncRNA 8244 by real-time RT-PCR assay. (**B**) PK-15 cells were no-infected or infected with SVA (MOI = 1.5). The mRNA expression of lncRNA 8244 at 3, 6, 9, 12, 24, 36 hpi. (**C**) The expression level of lncRNA 8244. (**D**) Expression of lncRNA 8244 in clinical samples of SVA-infected piglets. Total RNA was extracted to determine the relative mRNA expression of lncRNA 8244 with real-time RT-PCR assay. All data are represented as mean ± SD with three replicates. ** *p* < 0.01.

**Figure 2 microorganisms-11-00688-f002:**
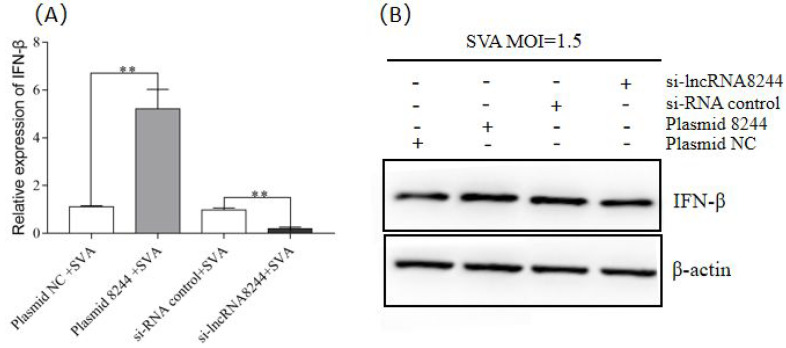
LncRNA 8244 can promote the expression of IFN-β. (**A**) PK-15 cells were treated with plasmid NC/plasmid 8244/iNC/si-lncRNA 8244 for 24 h, and then infected with SVA (MOI = 1.5) for an additional 24 h. The expression level of IFN-β was determined by real-time RT-PCR or Western blot (**B**). The mRNA level of IFN-β was normalized to the mRNA level of GAPDH, and anti-βactin was used as a control for Western blot sample loading. All data are represented as mean ± SD with three replicates. ** *p* < 0.01.

**Figure 3 microorganisms-11-00688-f003:**
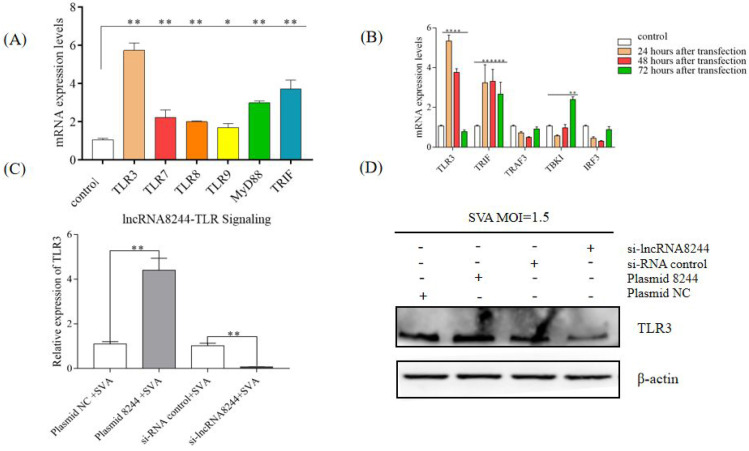
LncRNA 8244 can up-regulate the expression of key molecules in the TLR signaling pathway. (**A**) The expression of TLR3/TLR7/TLR8/TLR9/MyD88/TRIF. (**B**) The expression level of TLR3/TRIF/TRAF3/TBK1 and IRF3 was normalized to the mRNA level of GAPDH. (**C**) The expression level of TLR3 was determined by real-time RT-PCR assay or Western blot (**D**). The mRNA level of TLR3 was normalized to the mRNA level of GAPDH, and anti-βactin was used as a control for Western blot sample loading. All data are represented as mean ± SD with three replicates. * *p* < 0.05; ** *p* < 0.01.

**Figure 4 microorganisms-11-00688-f004:**
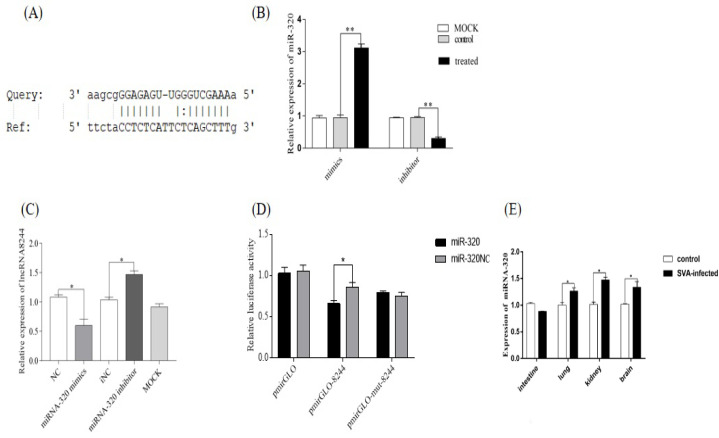
MiR-320 is the target gene of lncRNA 8244. (**A**) Predicted binding sites for miR-320 on the lncRNA 8244 transcript. (**B**) The expression level of miR-320 (**C**) The expression level of lncRNA 8244 after infected with SVA (MOI = 1.5) (**D**) Luciferase activity in PK-15 cells co-transfected with luciferase reporter containing pmirGLO-8244 or pmirGLO-mut-8244 and the mimics of miR-320. Data are presented as the relative ratio of Renilla luciferase activity and firefly luciferase activity. (**E**) The expression level of miR-320 in clinical samples of SVA-infected piglets. The mRNA level of TLR3 was normalized to the mRNA level of GAPDH, and anti-β-actin was used as a control for Western blot sample loading. All data are represented as mean ± SD with three replicates. * *p* < 0.05; ** *p* < 0.01.

**Figure 5 microorganisms-11-00688-f005:**
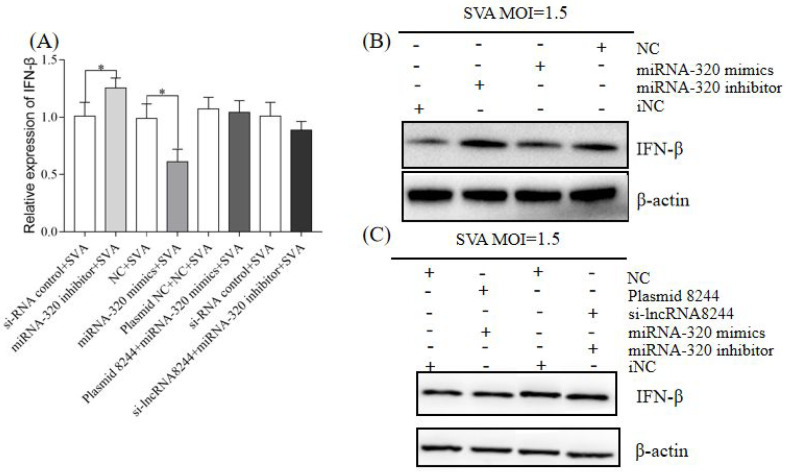
MiR-320 can suppress the expression of IFN-β (**A**) PK-15 cells were transfected or co-transfected with iNC/miRNA-320 inhibitor/miRNA-320 mimics/NC/plasmid NC and iNC/plasmid 8244 and ssc-miR-320 mimics/iNC and iNC/si-lncRNA 8244 and miR-320 inhibitor for 24 h, and infected with SVA (MOI = 1.5) for additional 24 h. The expression of IFN-βwas determined by real-time RT-PCR assay or Western blot (**B**,**C**). The mRNA level of IFN-β was normalized to mRNA level of GAPDH, and anti-βactin was used as a control for Western blot sample loading. All data are represented as mean ± SD with three replicates. *, *p* < 0.05.

**Figure 6 microorganisms-11-00688-f006:**
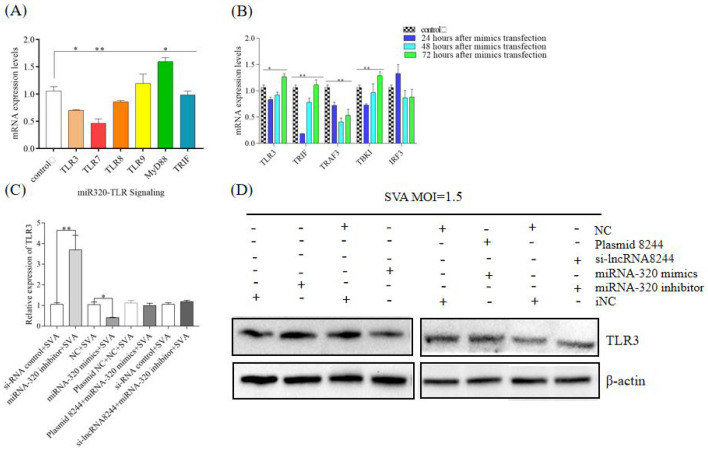
MiR-320 can down-regulate the expression of key molecules in the TLR signaling pathway. (**A**) The expression of TLR3/TLR7/TLR8/TLR9/MyD88/TRIF. (**B**) The expression of TLR3/TRIF/TRAF3/TBK1 and IRF3. (**C**) The expression of TLR3 was determined by real-time RT-PCR assay or Western blot (**D**). The mRNA level of TLR3 was normalized to the mRNA level of GAPDH, and anti-βactin was used as a control for Western blot sample loading. All data are represented as mean ± SD with three replicates. * *p* < 0.05; ** *p* < 0.01.

**Figure 7 microorganisms-11-00688-f007:**
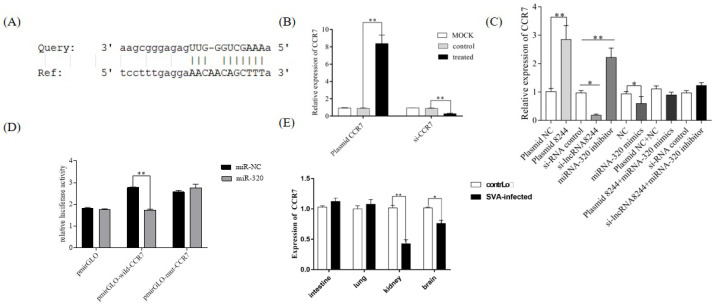
CCR7 is the target gene of miR-320. (**A**) Predicted binding sites for CCR7 on the miR-320. (**B**) PK-15 cells were treated with plasmid NC/plasmid CCR7/iNC/si-CCR7 for 24 h. The expression of CCR7 was determined by real-time RT-PCR assay and the mRNA level of CCR7 was normalized to the mRNA level of GAPDH. (**C**) The expression level of CCR7 was determined by real-time RT-PCR assay and the mRNA level of CCR7 was normalized to the mRNA level of GAPDH. (**D**) Luciferase activity in PK-15 cells co-transfected with luciferase reporter containing CCR7 or mutant and the mimics of miR-320. Data are presented as the relative ratio of Renilla luciferase activity and firefly luciferase activity. (**E**) Expression of CCR7 in clinical samples of SVA-infected piglets. Total RNA was extracted to determine the relative mRNA expression of CCR7 by real-time RT-PCR assay. All data are represented as mean ± SD with three replicates. * *p* < 0.05; ** *p* < 0.01.

**Figure 8 microorganisms-11-00688-f008:**
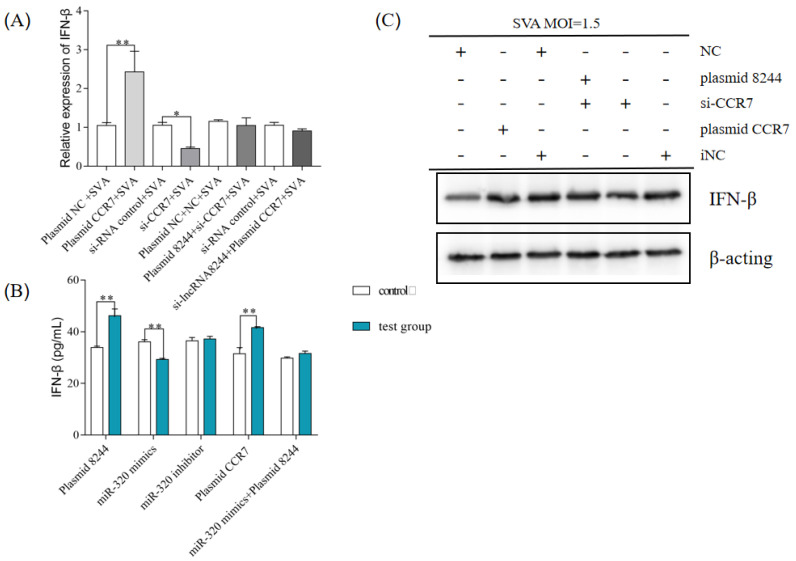
CCR7 can promote the expression of IFN-β (**A**) PK-15 cells were transfected or co-transfected with plasmid NC/plasmid CCR7/iNC/si-CCR7/plasmid NC and NC/plasmid 8244 and si-CCR7/iNC and iNC/si-lncRNA 8244 and plasmid CCR7 for 24 h. The expression level of IFN-β was determined by real-time RT-PCR assay and Western blot (**B**). The mRNA level of IFN-β was normalized to mRNA level of GAPDH or ELISA (**C**). All data are represented as mean ± SD with three replicates. * *p* < 0.05; ** *p* < 0.01.

**Figure 9 microorganisms-11-00688-f009:**
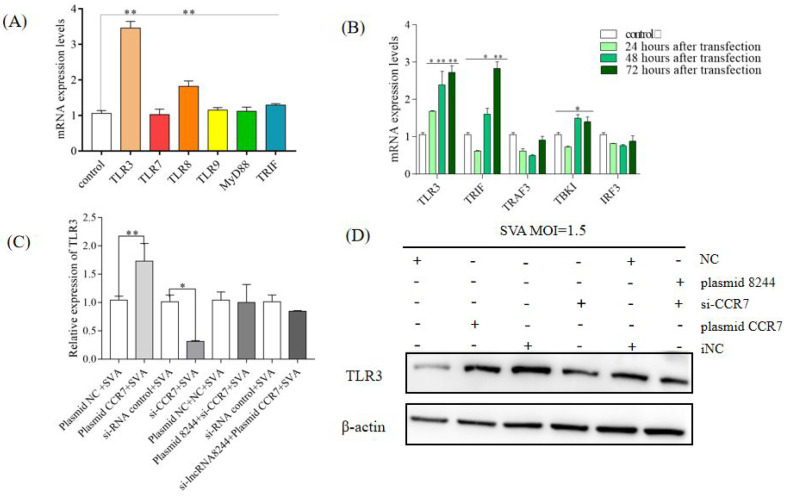
CCR7 can up-regulate the expression of key molecules in the TLR signaling pathway. (**A**) The expression level of TLR3/TLR7/TLR8/TLR9/MyD88/TRIF. (**B**) The expression level of TLR3/TRIF/TRAF3/TBK1 and IRF3. (**C**) The expression level of TLR3 was normalized to the mRNA level of GAPDH, and anti-βactin was used as a control for Western blot sample loading (**D**). All data are represented as mean ± SD with three replicates. * *p* < 0.05; ** *p* < 0.01.

**Table 1 microorganisms-11-00688-t001:** Primers used for real-time RT-PCR.

Primers	Sequence (5′-3′)	Amplification Length
IFN-β-F	GCTAACAAGTGCATCCTCCAAA	77 bp
IFN-β-R	AGCACATCATAGCTCATGGAAAGA	
IRF3-F	CCAGTGGTGCCTACACTCCT	191 bp
IRF3-R	AGAGGTGTCTGGCTCAGGAA	
TLR3-F	CGCCTCCTGGAAAACCAA	76 bp
TLR3-R	CCCTGAGTTGTCCTGCAACA	
TBK1-F	GCCTTTCTCGGGGTCTTCAA	74 bp
TBK1-R	ACACTTTTCCTGATCCGCCT	
TLR7-F	CGGTGTTTGTGATGACAGAC	134 bp
TLR7-R	AACTCCCACAGAGCCTCTTC	
TLR8-F	CACATTTGCCCGGTATCAAG	145 bp
TLR8-R	TGTGTCACTCCTGCTATTCG	
TLR9-F	GGCCTTCAGCTTCACCTTGG	151 bp
TLR9-R	GGTCAGCGGCACAAACTGAG	
MyD88-F	TGATGAACCGCAGGAT	438 bp
MyD88-R	ACTGTGCTACGGGCTGGATT	
TRIF-F	TGGGACATCCTTAGGGACATG	158 bp
TRIF-R	CCAGTGGACCTCAGGGAATG	
TRAF3-F	GTGTCAAGAAGGCATCG	164 bp
TRAF3-R	CCTCAAACTGGCAATCA	
CCR7-F	TGCTGGTGGTGGCTCTCCTTG	87 bp
CCR7-R	CCGTGGTGTTGTCGCCGATG	
LncRNA 8244-F	CAGAGGCAGGAACTGTGATGGC	145 bp
LncRNA 8244-R	GTGGTAGGTGAATCTGCGGAAGG	

## Data Availability

The original contributions presented in the study are included in the article. Further inquiries can be directed to the corresponding authors.

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
