# Peer review of "LncRNA 8244-ssc-miR-320-CCR7 Regulates IFN-β during SVA Infecting PK-15 Cells"

_microorganisms, 2023, doi:10.3390/microorganisms11030688_

Round 1

Reviewer 1 Report

Thank you for the opportunity to review this manuscript. In the manuscript, the authors studied how Seneca Valley infection can induce the production of IFN-b via TLR3. The results are interesting, however, due to there being missing information in the manuscript, it is not easy to understand. Also, for the discussion and conclusion, the inclusion of some controls and some results are needed.

Introduction:

1.     You are including results in the introduction.

2.     Page No.2 line 14: this information does not belong to the introduction.

3.     The last paragraph does not belong to the introduction.

4.     Based on references, information about lncRNA 8244 should be included. I understand the importance of lncRNA 8244 in the axis, but in the introduction is not clear why you start working with lncRNA 8244. 

Material and methods:

1.     Cell and viruses: you should include the cell line used. 

2.     Plasmids and antibodies: you worked with different plasmids, please write each plasmid and how it was designed and what it was designed for. Also, antibody names are needed.

3.     Include a subtitle with the information related to how the infection of the cells was conducted and another subtitle explaining the transfection.

4.     RT-PCR: as there are no references in Table 1, it means that the primers were designed for this research. In this sense, please included how the primers were designed. 

5.     Transduction and western blotting: the paragraph is written as a general protocol, step by step. Considering that the experiment is already done, please rewrite this paragraph. Also, the name of the antibodies used shall be included.

6.     Dual luciferase activity detection: Considering that the experiment is already done, please rewrite these paragraphs.

7.     As you included clinical samples, you should include here the number of samples and the precedence of the samples.

Results:

If the figures are showing the expression levels, why do you talk about up-regulation instead of overexpression?

Figure 1. 

1.     Figure 1B should be 1A and 1A should be 1B. 

2.     For Figures 1A, 1B, and 1C, what did you use as a control?

3.     Figure 1D what control did you use? lncRNA expression levels of non-infected piglets should be included.

Page 5, line 4.

1.     “plasmid 8244 was transfected” is not correct, you transfected cells with plasmids.

Figure 3.

1.     Figures 3A and B: what did you use as a control?

Page 6. 

1.     You should include the citation for the Software RegRNA 2.0

2.     You talk about a lncRNA 8244 mutant plasmid, which mutation did you do, what was the goal of that mutation?

Figure 4.

1.     Which control did you use and what was the difference between the control and MOCK?

2.     In the legend, you are mixing legend with methods

Figure 5.

1.     Figures 5B and C are not showing statistical differences, please check page 7, subtitle 3.4, lines 9-10.

Figure 7.

1.     You should include the citation for the Software RegRNA 2.0

You should unify the P<, sometimes it is in capital letters, sometimes in lower case, and sometimes in italics.

For the discussion, these results are needed

2.     How was the behavior of TLR3 in the tissues?

3.     Did you find RNA of the virus in the intestine? 

4.     If miRNA is a competitor of CCR7, why in the intestine could be their expression levels like the expression levels of the control?

5.     Should be a control (non-infected) per tissue.

Discussion:

Based on your results, in the tissues, the expression of lncRNA and CCR7 was lower or similar than in the control, while miRNA was overexpressed, meanwhile, in vitro, the behavior was the opposite. How can you explain this?

Cites:

You must include a space between the text and the citations throughout the manuscript.

Author Response

Point by point responses to the reviewer:

First, we would like to thank the reviewers and the editor for the positive and constructive comments and suggestions.

Comments and Suggestions for Authors

Thank you for the opportunity to review this manuscript. In the manuscript, the authors studied how Seneca Valley infection can induce the production of IFN-b via TLR3. The results are interesting, however, due to there being missing information in the manuscript, it is not easy to understand. Also, for the discussion and conclusion, the inclusion of some controls and some results are needed.

Thank you for your valuable comments, we have improved and revised these sections based on your suggestions. 

Introduction:

  1. You are including results in the introduction.

Answers: Thank you for your suggestion, we have revised the introduction section. Line 53-74

  1. Page No.2 line 14: this information does not belong to the introduction.

Answers: Thank you for your suggestion, we have revised the introduction section.

  1. The last paragraph does not belong to the introduction.

Answers: Thank you for your suggestion, we have revised the introduction section. Line 63-74

  1. Based on references, information about lncRNA 8244 should be included. I understand the importance of lncRNA 8244 in the axis, but in the introduction is not clear why you start working with lncRNA 8244. 

Answers: Thank you very much for your suggestion. Information about lncRNA8244 is not listed because it has not been studied by scholars so far. lncRNA8244 is among the lncRNAs that are significantly down-regulated in expression in our high-throughput sequencing results, so our attention is directed to it

Material and methods:

  1. Cell and viruses: you should include the cell line used. 

Answers: Thank you for your suggestions, we have added to the materials and methods section. Line 77-79

  1. Plasmids and antibodies: you worked with different plasmids, please write each plasmid and how it was designed and what it was designed for. Also, antibody names are needed.

Answers: Thank you for your suggestions, we have made changes. Line 83-89

  1. Include a subtitle with the information related to how the infection of the cells was conducted and another subtitle explaining the transfection.

Answers: Thanks to your suggestion, we have added a subheading related to how to perform cell infection and a subheading explaining transfection.  Line 100-104

  1. RT-PCR: as there are no references in Table 1, it means that the primers were designed for this research. In this sense, please included how the primers were designed. 

Answers: Thanks to your suggestion, we have made changes. Line 94

  1. Transduction and western blotting: the paragraph is written as a general protocol, step by step. Considering that the experiment is already done, please rewrite this paragraph. Also, the name of the antibodies used shall be included.

Answers: Thanks to your suggestion, we have rewritten it based on your suggestions. Line 106-116

  1. Dual luciferase activity detection: Considering that the experiment is already done, please rewrite these paragraphs.

Answers: Thanks to your suggestion, we have rewritten it based on your suggestions. Line 118-124

  1. As you included clinical samples, you should include here the number of samples and the precedence of the samples.

 Answers: Thanks to your suggestion, we have rewritten it based on your suggestions. Line 80-81

Results:

If the figures are showing the expression levels, why do you talk about up-regulation instead of overexpression?

Answers: Thank you for your suggestions. In our figures, the results are relatively quantitative, so we use the up-regulation.

Figure 1. 

  1. Figure 1B should be 1A and 1A should be 1B. 

Answers: Thank you for your suggestions, we have made changes.

  1. For Figures 1A, 1B, and 1C, what did you use as a control?

Answers: Thank you for your question, normally grown PK-15 cells were used as a control.

  1. Figure 1D what control did you use? lncRNA expression levels of non-infected piglets should be included.

Answers: Thank you for your suggestions. In our figures, the results are relatively quantitative.

Page 5, line 4.

  1. “plasmid 8244 was transfected” is not correct, you transfected cells with plasmids.

Answers: Thanks to your suggestion, we have rewritten it based on your suggestions.

Figure 3.

  1. Figures 3A and B: what did you use as a control?

Answers: Thanks to your suggestion, we have rewritten it based on your suggestions.

Page 6. 

  1. You should include the citation for the Software RegRNA 2.0

Answers: Thank you for your suggestions, we have made additions.

  1. You talk about a lncRNA 8244 mutant plasmid, which mutation did you do, what was the goal of that mutation?

Answers: The lncRNA 8244 mutant plasmid, which alters its binding site to miR-320. The purpose of the mutation was to verify its targeting relationship with miRNA-320 by a dual luciferase assay. We can provide mutation site information in the appendix.

Figure 4.

  1. Which control did you use and what was the difference between the control and MOCK?

Answers: Thank you for your question, normally grown PK-15 cells were used as a control.

  1. In the legend, you are mixing legend with methods

Answers: Thanks to your suggestion, we have rewritten it based on your suggestions.

Figure 5.

  1. Figures 5B and C are not showing statistical differences, please check page 7, subtitle 3.4, lines 9-10.

Answers: Thanks to your suggestion, we have rewritten it based on your suggestions.

Figure 7.

  1. You should include the citation for the Software RegRNA 2.0

 Answers: Thank you for your suggestions, we have made additions.

You should unify the P<, sometimes it is in capital letters, sometimes in lower case, and sometimes in italics.

 Answers: Thank you for your suggestion, we have made changes.

For the discussion, these results are needed

  1. How was the behavior of TLR3 in the tissues?

 Answers: Thank you for your suggestions, we have made additions (Supplement figure 1A).

  1. Did you find RNA of the virus in the intestine? 

Answers: Thank you for your question. SVA is detectable in the small intestine based on the toxicity of the tissue after SVA infection of piglets.

  1. If miRNA is a competitor of CCR7, why in the intestine could be their expression levels like the expression levels of the control?

Answers: Thank you for your question. Our interpretation is that the expression profiles of CCR7 as well as miR-320 were initially determined by high-throughput sequencing results of SVA-infected PK-15 cell, so there is some tissue variability. We have made changes in the manuscript

  1. Should be a control (non-infected) per tissue.

 Answers: Thank you for your suggestions, we have made additions (fig1,4,7).

Discussion:

Based on your results, in the tissues, the expression of lncRNA and CCR7 was lower or similar than in the control, while miRNA was overexpressed, meanwhile, in vitro, the behavior was the opposite. How can you explain this?

Answers: Thank you very much for your question, help us to add the shortage. In tissues, lncRNA and CCR7 expression was lower or similar than controls, whereas miRNA was overexpressed, which corresponds to our results at the cellular level. Here is an oversight on our part, only the expression levels after transfection of plasmids or RNAs are represented in the figure, and we have added to this (Supplement figure 1B).

Cites:

 You must include a space between the text and the citations throughout the manuscript.

Answers: Thank you for your suggestion, we have made changes

Reviewer 2 Report

This study investigated the role of LncRNA 8244 in the antiviral response during Senecavirus A virus infection. The authors chose to study LncRNA 8244 without providing background information or citation. Tang et al. found a competitive interaction between lncRNA 8244 with ssc-miR-320. Through this interaction, the researchers found that LncRNA 8244 can up-regulate the transcription of CCR7 and promote the expression of IFN-β. The results of this study provide insight into the relationship between the host's antiviral innate immune response and the endogenous RNA network, offering valuable information for the prevention and control of virus-related diseases and the early identification and intervention of severe cases. However, the manuscript could benefit from improved writing and more thorough data analysis to better present and interpret the findings.

In particular, this study lacks a critical component for a thorough analysis: a control without virus infection. Without a mock infection control, it is difficult to draw accurate conclusions about the specific role of LncRNA 8244 in the antiviral response during Senecavirus A infection. Additionally, the quality of the figures presented in the manuscript is inadequate, making it difficult to effectively interpret the data. This hinders the ability to draw meaningful conclusions from the study.

Furthermore, the statistical analysis in the study appears to be incorrect. The use of a paired sample t-test for multiple comparisons between multiple groups is not appropriate and could lead to inaccurate results.

To improve the quality of the manuscript and its findings, it is recommended to include a control without virus infection and to improve the quality of the figures. The low-quality western blots figures should be removed or put in a supplemental figure. The statistical analysis should be reviewed and corrected to ensure the validity of the conclusions drawn from the study. Additionally, all the long sentences in this draft should be re-write.

There are a few concerns listed below that would be important to address.

1.   In the materials and methods, this manuscript did not provide any information about the source and evaluation criteria of SVA-positive clinical samples. Moreover, this manuscript did not provide any information about the identification of the LncRNA 8244 in SVA-infected cells. A brief introduction or previously published reference should be made or cited here.

2.   Please modify all figures associated with clinical or animal-related data to be dot plots. Additionally, the study appears to have insufficient repetition of animal experiments (only three replicates), which is unacceptable in scientific research. It is important to repeat experiments multiple times to ensure the reliability and validity of the results.

3.   The difference between each group in the western blot figures is unclear. This makes it difficult to accurately interpret the results and draw meaningful conclusions from the study. Please use the intensity data from multiple replicates and plot them into new dot plots.

4.   The Figure 8 do not have panel D. But the author emphasized the date from Figure 8D.

5.   Please remove all the unnecessary figures or move them to supplementary figures, such as Figure 1C, Figure 4B, Figure 7B etc.

6.   The quality of the writing in the manuscript is not acceptable. There appear to be a large number of misspellings and grammar issues throughout the paper, which could hinder the ability of readers to understand the content and detract from the overall credibility of the study. 1.   It is recommended that the authors revise the English writing in the manuscript. To ensure that the writing is clear, concise, and free of errors, it is advised that the authors find a native English speaker or AI to proofread the manuscript.

Author Response

Point by point responses to the reviewer:

First, we would like to thank the reviewers and the editor for the positive and constructive comments and suggestions. Please find my itemized responses in below and my revisions in the re-submitted files.

Comments and Suggestions for Authors

This study investigated the role of LncRNA 8244 in the antiviral response during Senecavirus A virus infection. The authors chose to study LncRNA 8244 without providing background information or citation. Tang et al. found a competitive interaction between lncRNA 8244 with ssc-miR-320. Through this interaction, the researchers found that LncRNA 8244 can up-regulate the transcription of CCR7 and promote the expression of IFN-β. The results of this study provide insight into the relationship between the host's antiviral innate immune response and the endogenous RNA network, offering valuable information for the prevention and control of virus-related diseases and the early identification and intervention of severe cases. However, the manuscript could benefit from improved writing and more thorough data analysis to better present and interpret the findings.

In particular, this study lacks a critical component for a thorough analysis: a control without virus infection. Without a mock infection control, it is difficult to draw accurate conclusions about the specific role of LncRNA 8244 in the antiviral response during Senecavirus A infection. Additionally, the quality of the figures presented in the manuscript is inadequate, making it difficult to effectively interpret the data. This hinders the ability to draw meaningful conclusions from the study.

Furthermore, the statistical analysis in the study appears to be incorrect. The use of a paired sample t-test for multiple comparisons between multiple groups is not appropriate and could lead to inaccurate results.

To improve the quality of the manuscript and its findings, it is recommended to include a control without virus infection and to improve the quality of the figures. The low-quality western blots figures should be removed or put in a supplemental figure. The statistical analysis should be reviewed and corrected to ensure the validity of the conclusions drawn from the study. Additionally, all the long sentences in this draft should be re-write.

Answers: Thank you very much for your suggestions and we have made extensive changes to the manuscript based on your suggestions. First of all, we have re-run the statistical analysis and got the accurate results. Secondly, we have made changes in the manuscript for the image quality you mentioned.

There are a few concerns listed below that would be important to address.

  1. In the materials and methods, this manuscript did not provide any information about the source and evaluation criteria of SVA-positive clinical samples. Moreover, this manuscript did not provide any information about the identification of the LncRNA 8244 in SVA-infected cells. A brief introduction or previously published reference should be made or cited here.

Answers: Thank you for your suggestion. Information on clinical samples has been added to the manuscript(line 80-81). The expression of lncRNA 8244, miR-320 and CCR7 in cells has been supplemented in the Materials and Methods (Supplement figure 1). The background of lncRNA 8244 has not been studied in detail, and we will continue to follow up on this issue.

  1. Please modify all figures associated with clinical or animal-related data to be dot plots. Additionally, the study appears to have insufficient repetition of animal experiments (only three replicates), which is unacceptable in scientific research. It is important to repeat experiments multiple times to ensure the reliability and validity of the results.

Answers: Thank you for your suggestion. Combined with the nature of our experiment, the clinical samples are just to verify the expression of lncRNA 8244, miR-320 and CCR7. And in the last two years, clinical samples of SVA infection are really difficult to obtain. We will continue to follow up with samples if they are available in the future.

  1. The difference between each group in the western blot figures is unclear. This makes it difficult to accurately interpret the results and draw meaningful conclusions from the study. Please use the intensity data from multiple replicates and plot them into new dot plots.

Answers: Thank you for your suggestions. Based on your suggestion we grayed out the western blot results (Supplement figure 1).

  1. The Figure 8 do not have panel D. But the author emphasized the date from Figure 8D.

Answers: Thank you for the reminder. It was an oversight on our part and we have made corrections.  

  1. Please remove all the unnecessary figures or move them to supplementary figures, such as Figure 1C, Figure 4B, Figure 7B etc.

Answers: Thank you very much for your suggestion, we have made changes.

  1. The quality of the writing in the manuscript is not acceptable. There appear to be a large number of misspellings and grammar issues throughout the paper, which could hinder the ability of readers to understand the content and detract from the overall credibility of the study. 1.   It is recommended that the authors revise the English writing in the manuscript. To ensure that the writing is clear, concise, and free of errors, it is advised that the authors find a native English speaker or AI to proofread the manuscript.

Answers: Thank you very much for your advice, we are also aware of our shortcomings in English writing. We apologize for the poor language of our manuscript. We have now worked on both language and readability. We have also involed native english speaker and AI for language corrections. We really hope that the flow and language level have been substantially improved.

Round 2

Reviewer 1 Report

I would like to thank you for considering my suggestions. I could notice the changes.

The manuscript needs just a few corrections.

In subtitle 2.1, when you talk about two samples each, do you mean, healthy and unhealthy? I am wondering this because in the figures you talk about three replicates, so, do these replicates belong to technical o biological replicates?

In subtitle 2.2, the vendor of pmirGLO is missing. Also, information about anti-IFN-B is missing.

In subtitle 2.3, a reference for primers 5.0 is missing.

When I did a blast, I found that the primers that you used for amplifying LncRNA belong to the fatty acid desaturase 1 (FADS1), this should be mentioned in the manuscript.

In subtitle 2.4, please remove the space between 37 and C, and you should include 5% CO2.

In Subtitle 2.5, line 6, you should use blocked instead of closed. Line 9, you should use “were” instead of “are”

In subtitle 2.6, the first paragraph is still written as a protocol. Please, re-write the paragraph.

In figures 1A and 3A, X-axis, there is a square after control.

In the legends of the figures, the P value is written in italics, however in the text is written without italics, please unify this.

In subtitle 3.3, please, could you include the information about the mutation?

In subtitle 3.4, line 9, figures 5B and 5C, the western blots that you did are qualitative, they are not quantitative, which means that you cannot say if there was or was not a significant difference.

The supplementary material should be included in the text of the manuscript.

In the discussion, you should include what could happen with the small intestine, I mean, if SVA was detectable in the intestine and despite LncRNA was downregulated, why was the behavior of miR-320, CCR7, and TLR3 like the control?

Author Response

Point by point responses to the reviewer:

First, we would like to thank the reviewers and the editor for the positive and constructive comments and suggestions.

Comments and Suggestions for Authors

I would like to thank you for considering my suggestions. I could notice the changes.

The manuscript needs just a few corrections.

In subtitle 2.1, when you talk about two samples each, do you mean, healthy and unhealthy? I am wondering this because in the figures you talk about three replicates, so, do these replicates belong to technical o biological replicates?

Answers: Thank you for your question and I'm sorry for the confusion. The two repeats we are referring to here are two biological repeats. The three replicates in the figure refer to technical replicates. We performed three technical replicates for both biological replicates. In the figure we only show the results of the three technical replicates for one sample.

In subtitle 2.2, the vendor of pmirGLO is missing. Also, information about anti-IFN-B is missing.

Answers: Thank you for your reminder, we have added to the manuscript.

In subtitle 2.3, a reference for primers 5.0 is missing.

Answers: Thank you for your reminder, we have added to the manuscript.

When I did a blast, I found that the primers that you used for amplifying LncRNA belong to the fatty acid desaturase 1 (FADS1), this should be mentioned in the manuscript.Answers: Thank you for your reminder, we have added to the manuscript.You can view our genetic information on http://www.noncode.org

In subtitle 2.4, please remove the space between 37 and C, and you should include 5% CO2.

Thank you for your suggestions, we have made changes in the manuscript

In Subtitle 2.5, line 6, you should use blocked instead of closed. Line 9, you should use “were” instead of “are”.

Answers: Thank you for your suggestions, we have made changes.

In subtitle 2.6, the first paragraph is still written as a protocol. Please, re-write the paragraph.

Answers: Thank you for your suggestion, we have rewritten it.

In figures 1A and 3A, X-axis, there is a square after control.

Answers: Thank you for your suggestions, we have made changes.

In the legends of the figures, the P value is written in italics, however in the text is written without italics, please unify this.

Answers: Thank you for your suggestion, we have rewritten it.

In subtitle 3.3, please, could you include the information about the mutation?

Answers:Thank you for your suggestion, we have made changes to 3.4.

In subtitle 3.4, line 9, figures 5B and 5C, the western blots that you did are qualitative, they are not quantitative, which means that you cannot say if there was or was not a significant difference.

Answers: Thank you very much for your valuable advice. In response to your suggestion, we have performed a grayscale analysis of the western blot results and the results can indicate a significant difference. (Supplement figure 1)

The supplementary material should be included in the text of the manuscript.

Answers: Thank you for your suggestion, we have made the change.

In the discussion, you should include what could happen with the small intestine, I mean, if SVA was detectable in the intestine and despite LncRNA was downregulated, why was the behavior of miR-320, CCR7, and TLR3 like the control?

Answers: Thank you for your suggestion, we have made changes according to your suggestion

Reviewer 2 Report

The authors have made revisions to the manuscript, however, there are still a number of flaws that require further attention and correction.

1. The author did not provide an explanation for why a control group without viral infection was not included, nor was there a proper introduction or citation provided for the selection of LncRNA 8244.

2. Please revise the writing carefully. The paper appears to have a considerable number of misspellings (such as plasmid misspelled plasma) and grammar issues that need to be addressed through comprehensive editing.

Author Response

Point by point responses to the reviewer:

First, we would like to thank the reviewers and the editor for the positive and constructive comments and suggestions. Please find my itemized responses in below and my revisions in the re-submitted files.

Comments and Suggestions for Authors

The authors have made revisions to the manuscript, however, there are still a number of flaws that require further attention and correction.

  1. The author did not provide an explanation for why a control group without viral infection was not included, nor was there a proper introduction or citation provided for the selection of LncRNA 8244.

Answers: Thank you very much for your question. We did not design a control group without viral infection in the experiments of transfecting siRNA or expressing plasmids. The reason is to control the variables and get the results when only siRNA or expression plasmid is the variable. Our focus was on the effect of si-RNA, expression plasmid and control after SVA infection. We have chosen lncRNA8244 as the target of our study because lncRNA8244 was among the top five most significantly down-regulated genes expressed in high-throughput sequencing results after SVA infection of PK-15 cells.

  1. Please revise the writing carefully. The paper appears to have a considerable number of misspellings (such as plasmid misspelled plasma) and grammar issues that need to be addressed through comprehensive editing.

Answers: Thank you very much for your suggestion, we have checked the manuscript in detail and revised it. We have also revised the grammar.
